# Experimental Study on Migration and Intrusion Characteristics of Pulverized Coal in Propped Fractures

Qingao Zhu [1], Liming Yin [1], Qiming Huang [2,3,*], Enmao Wang [2] and Zhiguo Hou [2]

[1] College of Energy and Mining Engineering, Shandong University of Science and Technology, Qingdao 266590, China; 202182010027@sdust.edu.cn (Q.Z.); yinliming@sdust.edu.cn (L.Y.)

[2] College of Safety and Environmental Engineering, Shandong University of Science and Technology, Qingdao 266590, China; 17854257478@163.com (E.W.); hzg3643@163.com (Z.H.)

[3] Mine Disaster Prevention and Control-Ministry of State Key Laboratory Breeding Base, Shandong University of Science and Technology, Qingdao 266590, China

[*] Correspondence: skdhuang@163.com

**Abstract:** Aiming at the problem of pulverized migration and plugging propped fractures during coal seam fracturing, we experimentally studied the pressure changes and pulverized coal blocking characteristics with deionized water and solutions of three surfactants including 1227 ($C_{21}H_{38}ClN$), SDS ($C_{12}H_{25}SO_4Na$) and TX-100 ($C_{34}H_{62}O_{11}$). A device capable of visualizing propped fractures was established, and simulation experiments were conducted with solutions of different surfactants at different injection flow rates. The obtained images were binarized and analyzed to quantify the pulverized coal blockage degrees of facture under different conditions. The experimental results show that: (1) The higher the injection flow rate, the higher the inlet pressure. (2) All three surfactants can lower the injection pressure, as compared with water alone. SDS decreases the injection pressure more obviously at low injection flow rates, and the other two perform better at high injection flow rates. (3) Similar to their effects on inlet pressure, the ratio of pulverized coal in SDS solution is lower at low injection flow rates, while TX-100 and 1227 solutions show lower ratios of pulverized coal at high injection flow rates. Our work has provided a theoretical support for coal blockage removal and pressure reduction in propped fractures during coal seam fracturing to improve coal seam permeability and further improves the dust prevention effect of coal seam water injection.

**Keywords:** propped fracture; coal seam water injection; fracture seepage; fracture plugging

## 1. Introduction

The problem of dust pollution is becoming more and more serious, with the improvement of coal mine mechanization seriously affecting the health of underground equipment and personnel. Coal seam water injection can effectively prevent and control dynamic disasters in coal mining, such as mine dust and coal and gas outbursts [1,2]. With the increase of mining depth, coal seam water injection gradually becomes more difficult [3]. For the fracturing of some deeply buried coal seams with poor permeability, proppants are added into fracturing fluids to prevent fracture closure. After a proppant is added, the early fracturing and the friction and extrusion of the proppant on the fracture wall produce pulverized coals that migrate in the propped fracture with the fluid and eventually block the fracture [4], causing low coal seam permeability, low fracture conductivity and poor water injection effects. Therefore, reasonable prevention and control measures of pulverized coals are very important to achieve good permeability and fracture conductivity of coal seams. A large number of experiments on pulverized coal prevention and control have been conducted in the lab and by field study [5–7]. It is found that the surface wetting properties of pulverized coal can be greatly changed by the introduction of surfactants [8–18] or changing its particle size and other variables, providing a theoretical basis for pulverized coal control [19–21]. Some scholars also studied pulverized coal control from

the perspective of its migration in propped fracture by simulating its intrusion process into propped fracture in the lab. The migration law and the damaging mechanism to fracture conductivity were investigated by varying the particle size of pulverized coal, fracture width, injection flow rate and proppant [22–26]. In addition, the permeability of pulverized coal and blocked coal seams is also quantitatively described using mathematical models [27–30]. These extensive studies provide understanding of the properties of pulverized coal and surfactant and the fracture conductivity during the migration of pulverized coal. Yet, discussion on the injection pressure changes caused by the mixing of pulverized coal and proppant and the actual pulverized coal blockage state of propped fracture is still lacking. In particular, the migration of pulverized coal in propped fracture remains unclear from the perspective of injection pressure, which makes it difficult to affectively improve pulverized coal prevention and control in coal seam water injection. In this work, a certain amount of pulverized coal was mixed into propped fractures, and the solutions of different surfactants were injected into the fractures at different injection flow rates. The pressure changes and the actual blocking statuses under different injection conditions were recorded to reveal the pulverized coal prevention and control effects of the surfactants in coal seam water injection and to provide a scientific basis for the actual coal mine production.

## 2. Experimental Design

### 2.1. Experimental Setup

A pulverized coal migration visualization and fracture simulation device capable of generating fractures with different opening degrees was developed in-house. It was mainly composed of a TYD-01 syringe pump (produced by Baoding Lead Fluid Technology Co., Ltd., Baoding, China), pressure monitoring system (which consists of the Microfluidic Sensor Reader sensor data acquisition card, the PS3 model liquid pressure sensor and the Elveflow Smart Interface v3.0.19 acquisition software, which are all produced by Microfluidics Company Elvesys Group from France, acquisition frequency, 20 per second), a stereo microscope (produced by Leica AG in Germany, the S9I has a 10 million pixel integrated lens with a maximum resolution of 500 lp/mm), a computer (HP 290 G3 SFF Business PC produced by Hewlett Packard Inc., Palo Alto, CA, USA), glass plates and a holder. The solution in the microfluidic pump was pumped into the pre-laid visible fracture where pulverized coal was mixed with the proppant. The pulverized coal migrations in the propped fracture were visualized, and injection pressure changes were recorded as different solutions were injected at different flow rates.

### 2.2. Coal and Solution Samples

To explore the effects of different surfactants and injection flow rates on injection pressure, as well as the migration law of pulverized coal in propped fractures, three injection flow rates and four solutions were designed. Deionized water and the solutions of 1227 ($C_{21}H_{38}ClN$), SDS ($C_{12}H_{25}SO_4Na$) and TX-100 ($C_{34}H_{62}O_{11}$) and the flow rates of 4 mL/min, 7 mL/min and 10 mL/min composed a total of 12 groups of tests. Coal powder with particle sizes >120 mesh and white quartz sand with particle sizes of 16–20 mesh (0.85–1.18 mm) were used as the pulverized coal sample and the proppant. The sanding concentration was 0.5 g/cm$^2$. The selection of pulverized coal particle size is based on the particle size of pulverized coal produced in CBM drainage. The maximum mass ratio is 0.1–150 μm, which accounts for 0.64–3.71%. Therefore, pulverized coal with a particle size less than 125 μm is selected for the test [31]. The actual water injection flow rate is used to simulate the water injection flow rate in the test. Three flow rates are set to represent the flow rate entering the fracture, which are 4 mL/min, 7 mL/min, and 10 mL/min, respectively. The corresponding field flow rates are 1 m$^3$/h, 1.7 m$^3$/h and 2.5 m$^3$/h. The sand concentration in engineering practice is judged according to the situation on site. Most of the sand concentration is 5, 7.5, 10 and 15 kg/m$^2$. We choose the commonly used sand concentration of 5 kg/m$^2$, which is converted to 0.5 g/cm$^2$. (The volume density of the quartz sand selected in this experiment is 1.426 g/cm$^3$, and the simulated crack size

is 3 cm × 2 cm × 0.35 cm = 2.1 cm$^3$. Therefore, the crack volume and volume density are calculated: 1.426 g /cm$^3$ × 2.1 cm$^3$ = 2.99 g. The sand concentration is 0.5 g/cm$^2$. The sand area is 6 cm$^2$, and the quartz sand in the crack is 6 cm$^2$ × 0.5 g/cm$^2$ = 3 g.) The information of surfactants is shown in the following Table 1.

**Table 1.** Information on surfactants.

| English Synonym | Formula | CAS No. | Manufacturer | Purity | Molecular Weight |
|---|---|---|---|---|---|
| SDS | $C_{12}H_{25}SO_4Na$ | 151-21-3 | Shanghai Macklin Biochemical Co., Ltd., Shanghai, China | AR, 92.5–100.5% | 288.38 |
| 1227 | $C_{21}H_{38}ClN$ | 139-07-1 | Shanghai Macklin Biochemical Co., Ltd., Shanghai, China | AR, 99% | 339.99 |
| TX-100 | $C_{34}H_{62}O_{11}$ | 9002-93-1 | Beijing Solarbio Science & Technology Co., Ltd., Beijing, China | >98% | 647 |

*2.3. Solution Surface Tension Test*

Surface tension is a special force and a manifestation of liquid properties. The surface tension of different liquids is different, which is an important physical parameter to discuss the liquid surface and its properties. Whether the liquid can infiltrate the solid is related to its surface tension. The smaller the value, the easier it is to infiltrate the solid. This experiment uses the Kruss DSA30 droplet shape analyzer (produced by Kruss, Hamburg, Germany). The Table 2 shows the surface tension of the solution used. It can be seen that the surface tension of different solutions is different, and the surface tension decreases with the increase in concentration. The surface tension of the three surfactants is the smallest when the mass concentration is 1%. Therefore, in order to avoid the experimental error, a solution of 1% mass concentration is selected for the test. It can be seen from the table that the surface tension of 1227 solution is the smallest at 1% mass concentration, and the surface tension of SDS solution is the largest at 0.1% mass concentration. The measured surface tension of deionized water is 71.7 mN·m$^{-1}$.

**Table 2.** Surface tension coefficient of surfactant.

| wt% | 0.1 | 0.25 | 0.5 | 0.75 | 1 |
|---|---|---|---|---|---|
| Surface tension of 1% SDS solution/mN·m$^{-1}$ | 40.98 | 34.12 | 33.66 | 32.58 | 32.49 |
| Surface tension of 1% 1227 solution/mN·m$^{-1}$ | 32.01 | 31.64 | 31.96 | 31.96 | 31.78 |
| Surface tension of 1% TX-100 solution/mN·m$^{-1}$ | 39.35 | 36.85 | 36.6 | 36.65 | 36.6 |

*2.4. Methods*

Before the experiment, check whether the value of the pressure sensor is normal, set it to zero, connect each pipeline and use the micro-flow pump to inject 4, 7, 10 and 12 mL/min flow rate of liquid to observe the tightness of the crack and whether there is water seepage at the edge. The mixture of coal powder and proppant was put into the fracture between glass plates fixed with a holder. As shown in Figure 1, the microfluidic pump, pressure sensor, glass fracture, stereo microscope and computer were connected in order. The injection flow rate was set as shown in Table 1. The experiment was started once the sensor record was stable. The specific experimental design is shown in Table 3.

**Table 3.** Visual propped fracture experimental design.

| No. | Fluid | Coal Powder (g) | Particle Size of Proppant (Mesh) | Particle Size of Coal Powder (Mesh) | Flow Rate (mL/min) | Time (min) |
|---|---|---|---|---|---|---|
| 1 | water | 0.2 | 16–20 | >120 | 4 | 40 |
| 2 | water | 0.2 | 16–20 | >120 | 7 | 40 |
| 3 | water | 0.2 | 16–20 | >120 | 10 | 40 |
| 4 | 1% SDS | 0.2 | 16–20 | >120 | 4 | 40 |
| 5 | 1% SDS | 0.2 | 16–20 | >120 | 7 | 40 |
| 6 | 1% SDS | 0.2 | 16–20 | >120 | 10 | 40 |
| 7 | 1% 1227 | 0.2 | 16–20 | >120 | 4 | 40 |
| 8 | 1% 1227 | 0.2 | 16–20 | >120 | 7 | 40 |
| 9 | 1% 1227 | 0.2 | 16–20 | >120 | 10 | 40 |
| 10 | 1% TX-100 | 0.2 | 16–20 | >120 | 4 | 40 |
| 11 | 1% TX-100 | 0.2 | 16–20 | >120 | 7 | 40 |
| 12 | 1% TX-100 | 0.2 | 16–20 | >120 | 10 | 40 |

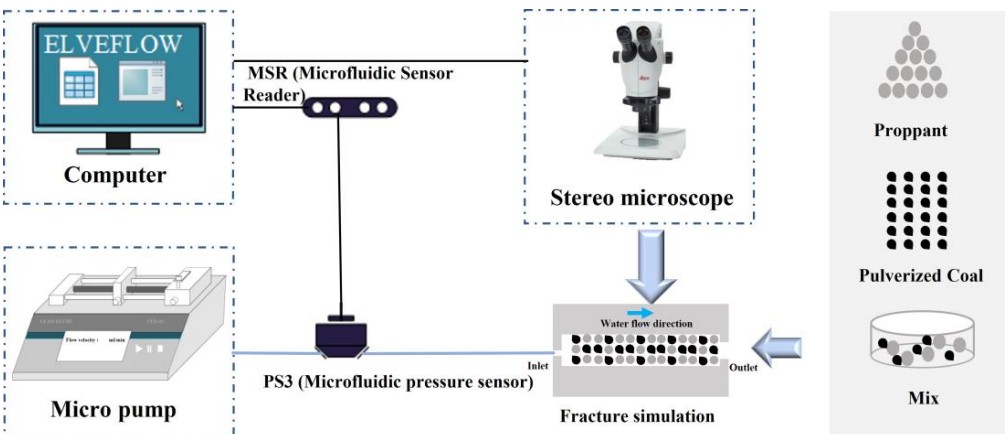

**Figure 1.** Experimental flow chart.

## 3. Results and Analysis

### 3.1. Injection Pressure Changes of Different Surfactant Solutions at Different Flow Rates

A series of propped fracture experiments were conducted with the solutions of three different surfactants using water was used as the control group. The pressure changes at different flow rates and those of different surfactant solutions at the same flow rate were analyzed and compared.

Figure 2 shows the injection pressure changes during the injection of deionized water at different flow rates. As can be seen, the inlet pressure gradually increases with the increase of the flow rate [6]. At the same injection flow rate, the inlet pressure gradually becomes stable with the prolongation of injection time, and the coal powder migration reaches the dynamic equilibrium in the propped fracture. The inlet pressures at the injection flow rates of 4 mL/min, 7 mL/min and 10 mL/min were, respectively, averaged and used as the reference pressures for the injections of the solutions of surfactants.

Figure 3a reveals that the inlet pressure gradually increases with the increase of the injection flow rate of 1% SDS solution. Compared with that of deionized water injection, the mean inlet pressure decreases by 14.2% at the flow rate of 4 mL/min, increases by 2.7% at the flow rate of 7 mL/min and drops by 3.1% at the flow rate of 10 mL/min. The SDS solution reduces the injection pressure most at the flow rate of 4 mL/min because the contact between the anionic wetting agent and the coal powder makes the coal surface prone to being combined with water [32,33]. More channels are formed from the coal powder migrating along with the injection flow, leading to the injection pressure drops. Further increasing the injection flow rate only changes the pressure growth rate slightly.

On the other hand, the reason is that the surface tension of the SDS solution is reduced, and it is easier to wet pulverized coal [34].

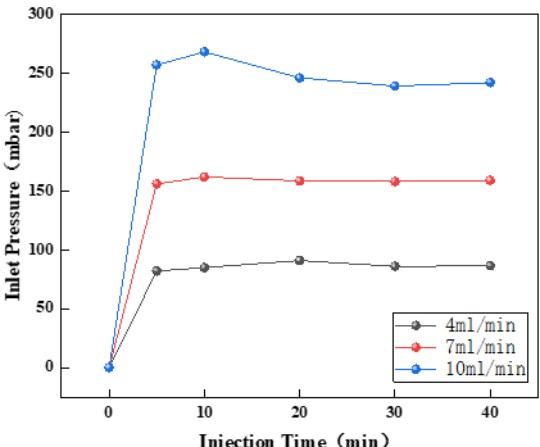

**Figure 2.** Pressure versus time curves of water injected at different flow rates.

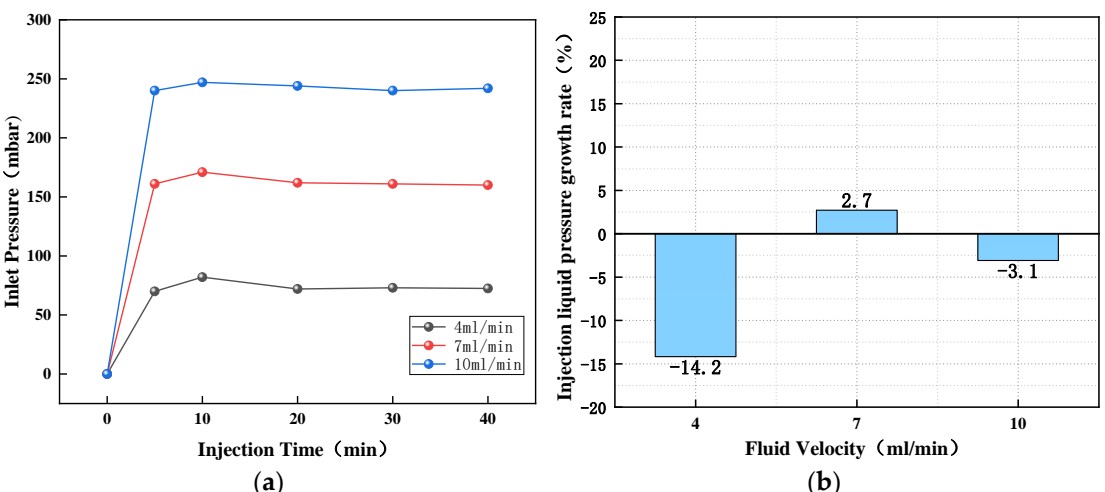

**Figure 3.** Pressure–time curves and pressure growth rates for the injection of 1% SDS at different flow rates. (**a**) Pressure curves at different flow rates; (**b**) Pressure growth rate.

As shown in Figure 4, the inlet pressure also gradually increases with the increase of injection flow rate of 1% 1227. Compared with that of deionized water injection, the inlet pressure shows no significant changes at the injection flow rate of 4 mL/min, drops by 6.5% at the injection flow rate of 7 mL/min and decreases by 8.9% at the injection flow rate of 10 mL/min. The cationic surfactant 1227 forms an unstable hydrophobic layer after contacting the surface of pulverized coal, which easily leads to the aggregation of pulverized coal [34]. Therefore, the pressure growth rate decreases with the prolongation of injection time.

Similarly, the injection pressure of 1% TX-100 solution gradually increases with the increase of injection flow rate (Figure 5). Compared with that of deionized water injection, the average injection pressure of 1% TX-100 increases by 21.5% at the injection flow rate of 4 mL/min, decreases by 4.85% at the injection flow rate of 7 mL/min and drops by 6.2% at the injection flow rate of 10 mL/min. Figures 3a, 4a and 5a of the pressure curve of the peak represents the pressure reaching its maximum value. It shows that at this time, the pulverized coal in the crack flows into each flow channel with the flow of the liquid and causes blockage, so the pressure peak appears. In addition to the decrease in surface tension of the surfactant, the pulverized coal is easy to wet. Another reason is that the surface of the pulverized coal and the anionic surfactant contains both hydrophobic

and hydrophilic groups. In the anionic surfactant solution, the hydrophobic group on the surface of the pulverized coal is adsorbed with the hydrophobic group of the anionic surfactant, and the hydrophilic group of the anionic surfactant faces the liquid, making the pulverized coal easy to wet. The cationic surfactant adsorbs the positively charged hydrophilic group on the surface of the pulverized coal in the solution, and the negatively charged hydrophobic group is also adsorbed on the surface of the pulverized coal due to the van der Waals force, forming a hydrophobic layer and reducing the aggregation of the pulverized coal. The adsorption mechanism of non-ionic surfactants on coal dust is different from that of ionic surfactants. Non-ionic surfactants do not have an ionization phenomenon in water but exist in solution in molecular form. The hydrophobic groups of non-ionic surfactants are adsorbed on the surface of pulverized coal, and the hydrophilic groups face the liquid. These hydrophilic groups will accelerate wetting [34–37]. In all, different types of surfactants show different performances in the injection pressure reduction.

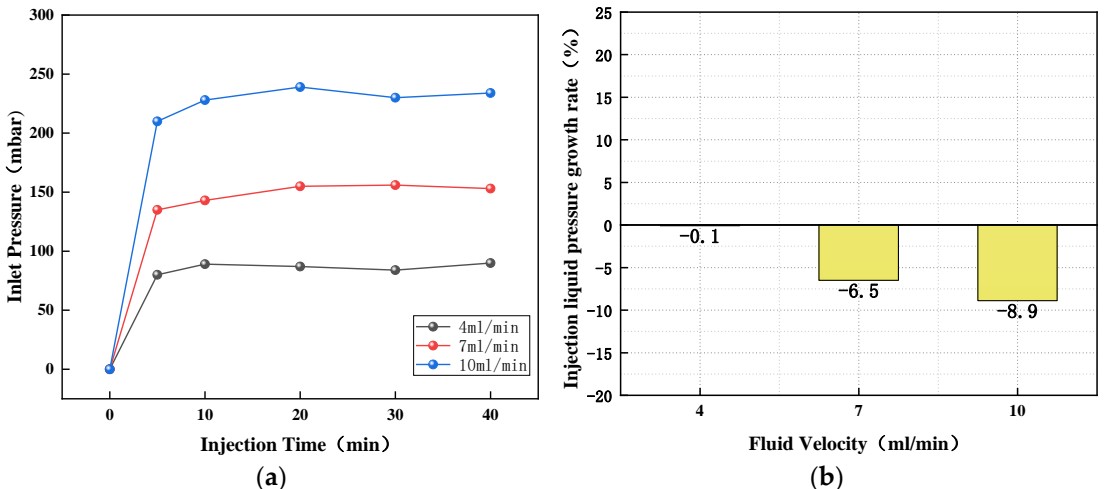

**Figure 4.** Pressure–time curves and pressure growth rates for the injection of 1% 1227 at different flow rates. (**a**) Pressure curves at different flow rates; (**b**) Pressure growth rate.

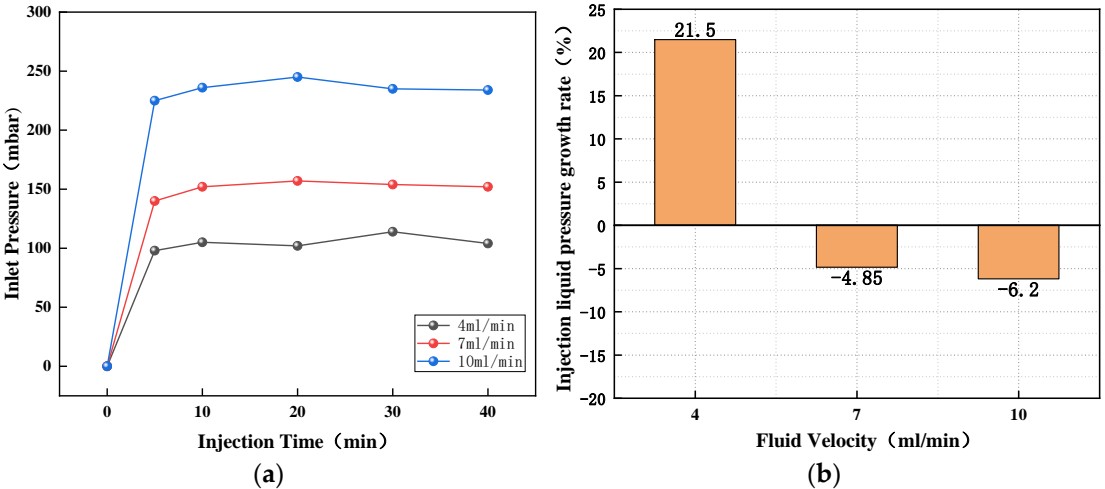

**Figure 5.** Pressure–time curves and pressure growth rates for the injection of 1% TX-100 at different flow rates. (**a**) Pressure curves at different flow rates; (**b**) Pressure growth rate.

### 3.2. Images Obtained with Solutions of Different Surfactants at Different Flow Rates

Pulverized coal migrates and flows with water or surfactant solution in propped fractures. When the migration of some coal particles is blocked by pore cracks, they deposit and block the fracture. As shown in Figure 6, These cracks that intercept pulverized coal

are shrinking pore throat channels. Due to the shrinkage of the width of the channel, the pulverized coal forms a bridging blockage at the pore throat. In addition, there is another effect that causes blockage in the channel of the crack. When the bubble flows to the narrow mouth of the crack, the diameter of the bubble is larger than the diameter of the hole, so that the pulverized coal cannot pass through and form a blockage [7,38]. Other coal particles move forward with the flow. The coal deposition can block the migration channels of pulverized coal in propped fractures.

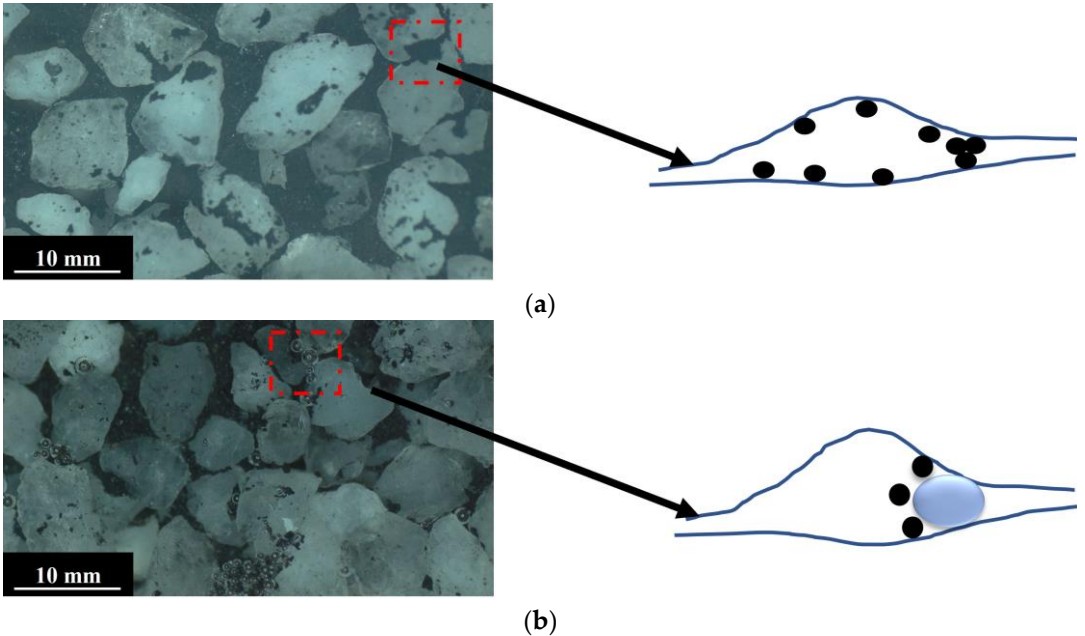

**Figure 6.** Two kinds of blocking diagram. (**a**) Bridge blockage; (**b**) Bubble embolization.

In this experiment, the gap between glass plates was designed to simulate propped fracture, and white quartz sand was used as the proppant. The deposition of coal powder can be used to simulate the blockage of propped fracture. After 40 min of the experiment, the image of the propped fracture was recorded and binarized for further analysis. Because of the obvious black-and-white contrast between pulverized coal and proppant, MATLAB (R2020b) software is used to binarize the image. The principle of image binarization is to convert an RGB image into a gray image, automatically select a threshold value and then convert it into a binary image. The operation steps are realized by the functions in the MATLAB software. The RGB image is imported into the software, and the gray thresh function is used to convert the gray image, and the threshold is automatically determined. The im2bw function is converted into a binary image. The pixels on the binary image have only two values of black and white (0 means black; 1 means white). Figure 7 is a schematic diagram of converting an RGB image into a binary image. On the binarized image, the pixels with gray level of 0 (black) refer to the coal particles. The ratio of the number of pixel 0 (black) to the total number of pixels represents the ratio of pulverized coal (R). R can be expressed as Equation (1).

$$R = P/G, \tag{1}$$

where R is the portion of pulverized coal, P is the number of pixels with gray level of 0 and G is the total number of pixels.

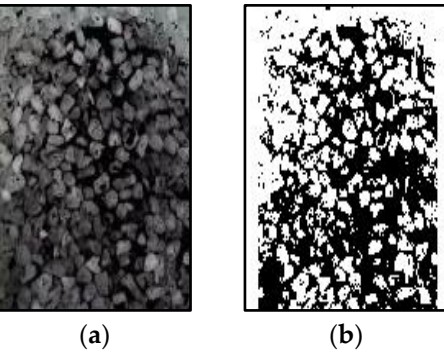

**Figure 7.** Binarization of original image. (**a**) Original image; (**b**) Binarized image.

The ratios of pulverized coal in different solutions injected at different flow rates were obtained, which are combined with the inlet pressure to reveal the coal find migration characteristics with different surfactants added and at different injection flow rates.

### 3.2.1. Ratios of Pulverized Coal at Different Flow Rates

Figure 8 shows the ratios of pulverized coal in different solutions injected at the flow rate of 4 mL/min. As can be seen, the R in the SDS solution is the smallest with a value of 30.4% at this low flow rate, indicating that it has the highest fluidity. As demonstrated above, the SDS solution decreases the injection pressure by 14.2% at this flow rate (Figure 3). These results suggest that the solution discharges more pulverized coal, increases the number of seepage channels and thus lowers the inlet pressure.

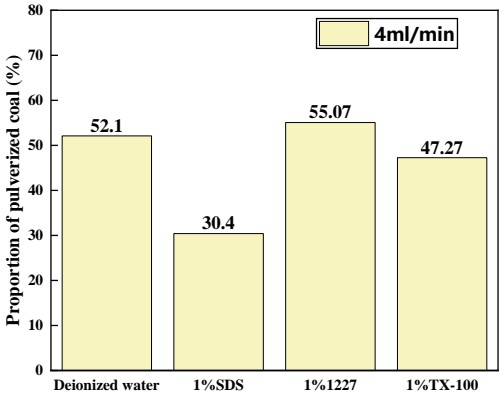

**Figure 8.** Ratios of pulverized coal in the different solutions injected at 4 mL/min.

As the injection flow rate increased to 7 mL/min, the R in TX-100 solution becomes the smallest with the value of 15.98% (Figure 9), indicating that the injection of TX-100 solution at this flow rate causes the least fracture blockage. Despite its low R, the TX-100 solution only slightly decreases the injection pressure because the interaction between this non-ionic surfactant with pulverized coal increases the viscosity of the solution (Figure 5).

At the injection flow rate of 10 mL/min, both the 1227 and TX-100 solutions show significantly lower R, with the values of 5% and 6.76%, respectively (Figure 10). As shown in Figures 4 and 5, the 1227 and TX-100 solutions lower the injection pressure by 8.9% and 6.2%, respectively, at this flow rate. These results indicate that both non-ionic and cationic surfactants perform very well on reducing pulverized coal blockage in propped fracture at high injection flow rates.

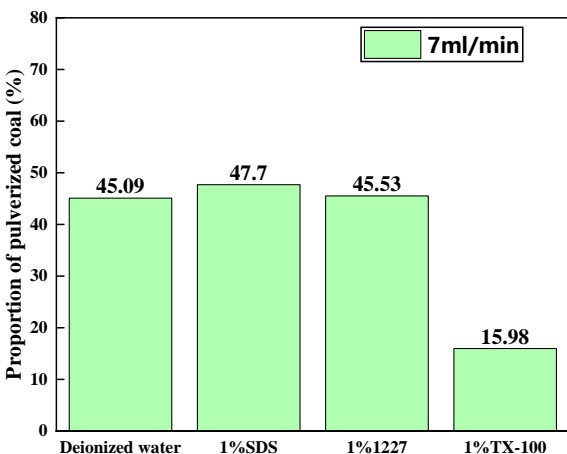

**Figure 9.** Ratios of pulverized coal in the different solutions injected at 7 mL/min.

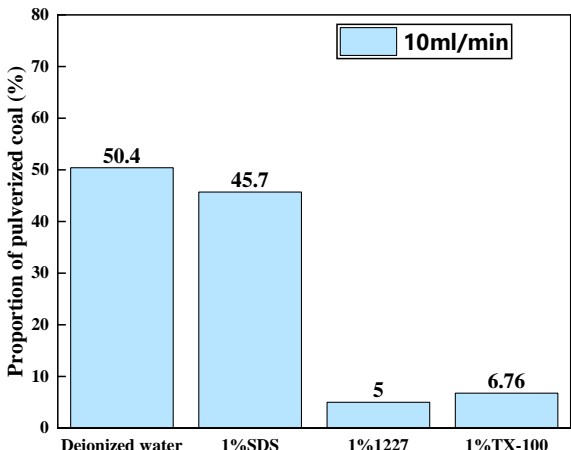

**Figure 10.** Ratios of pulverized coal in the different solutions injected at 10 mL/min.

The comparison of the ratios of pulverized coal in different solutions injected at different flow rates suggests that all the surfactant solutions can improve the pulverized coal blockage in propped fracture but show different ratios of pulverized coal at low, medium and high injection flow rates. Both R and injection pressure of the anionic surfactant solution are lower at low injection flow rates.

3.2.2. Ratios of Pulverized Coal in Solutions of Different Surfactants

Figure 11 shows the ratios of pulverized coal in the solutions of different surfactants injected at different flow rates. As can be seen, the R of deionized water remains almost constant as the injection flow rate varied. The performance of the SDS solution at the low flow rate is the best among the four fluids, while the TX-100 and 1227 solutions perform better at the high flow rate with lower R and thus cause less blockage. The TX-100 solution also works very well at the medium flow rate with an R of 15.98%. Overall, the TX-100 solution can be applied to a wider range of flow rate. From the perspective of a single solution, the R of the SDS solution increases with the increase of flow rate, and those TX-100 and 1227 solutions decrease with the increase of flow rate. We analyze the reason for this phenomenon because, at a medium-high injection speed, most of the pulverized coal is quickly pushed from the inlet end to the outlet end by the solution. Due to the small width of the outlet end, a large amount of pulverized coal accumulates at the outlet end. The hydrophobic group of the nonionic surfactant (TX-100) [39] is adsorbed on the surface of the pulverized coal, and the hydrophilic group faces the liquid. These hydrophilic groups will accelerate wetting. Due to the accumulation of pulverized coal in the pore throat channel, the solution gradually wets the pulverized coal at the blockage. At the same time,

due to the flow rate and pressure, the pulverized coal is quickly discharged, resulting in a rapid decrease in the amount of pulverized coal. The pulverized coal accumulated behind it is not fully wetted and discharged. The wetting mechanism of pulverized coal at low speed is consistent with that at medium and high speed, but the migration of pulverized coal at low speed is not as fast as that at medium and high speed, and there is enough time to wet pulverized coal, which leads to the solution being able to find a lot of seepage channels in the fracture and take away some wet pulverized coal, so that the content of pulverized coal in the fracture is higher and then tends to be stable [40]. The wetting effect of anionic surfactant (SDS) and cationic (1227) and non-ionic (TX-100) surfactants on the proportion of pulverized coal at each flow rate is better at each flow rate because the wetting rate of anionic surfactant is faster when contacting pulverized coal and the adsorption of anionic surfactant is stronger, so there is no difference between cationic (1227) and non-ionic (TX-100) surfactants at medium and high flow rates.

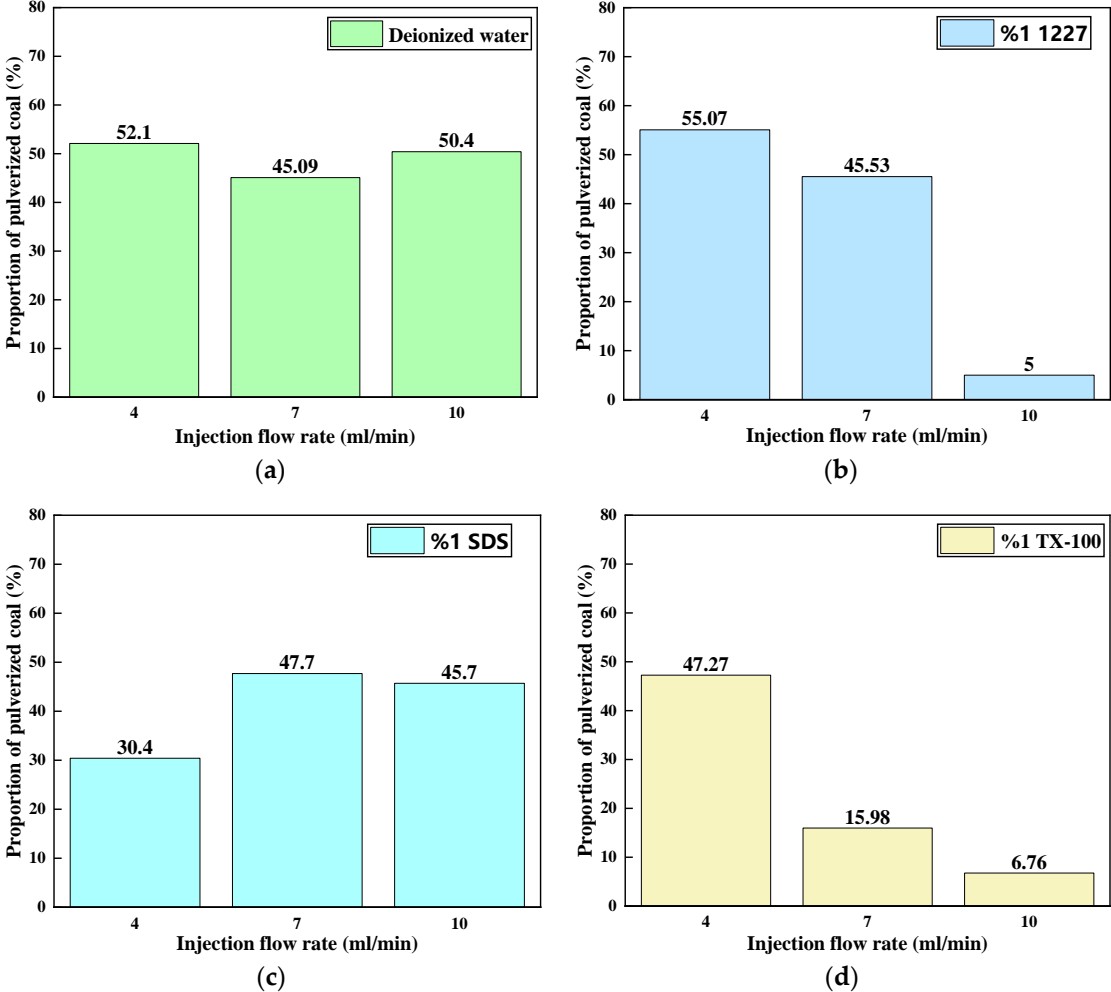

**Figure 11.** Ratios of pulverized coal in solutions of different surfactants at different injection flow rates. (**a**) The proportion of pulverized coal in deionized water. (**b**) The proportion of pulverized coal in 1% 1227. (**c**) The proportion of pulverized coal in 1% SDS. (**d**) The proportion of pulverized coal in 1% TX-100.

### 3.3. Formula Derivation and Experimental Verification of Inlet Pressure

If the channels formed in proppant fractures are considered as a capillary bundle model, the inlet pressure equation can be deduced with the Carman–Kozeny equation for the relationship between the seepage channel r and permeability (K) and the absolute permeability equation as follows.

The absolute permeability equation can be expressed as:

$$Q = \frac{K \cdot \Delta P \cdot A}{\mu L} \tag{2}$$

where Q is the flow rate, K is the permeability, $\Delta P$ is the pressure difference, $A$ is the area, $L$ is the length and $\mu$ is the viscosity.

The Carman–Kozeny [41,42] equation can describe the relation between seepage change ($r$) and K as:

$$K = \frac{\varphi r^2}{8\tau^2} \tag{3}$$

where $r$ is the effective flow radius, $\tau$ is the tortuosity of the effective flow pores and $\varphi$ is the porosity. The porosity can be obtained from the capillary bundle model as:

$$\varphi = \frac{N\pi r^2 L}{AL} \tag{4}$$

where $A$ is the cross-sectional area of the unit, $L$ is the length of the unit, $N$ is the number of capillaries in the unit and $r$ is the effective radius.

As shown in Figure 12, the entire section can be divided with an equilateral triangle with side length $R$, and the number of capillaries can be calculated as:

$$N = \frac{A}{\sqrt{3}R^2} \tag{5}$$

where $R$ is the proppant particle radius, and $A$ is the unit volume area.

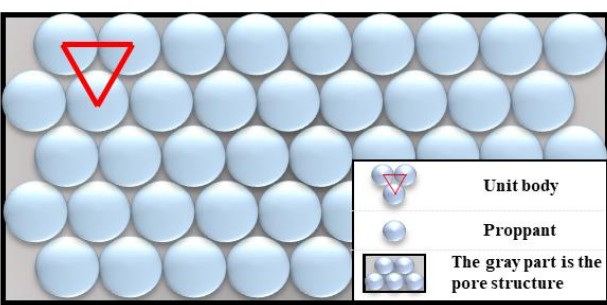

**Figure 12.** Capillary bundle unit.

The inlet pressure then can be obtained from Equations (2)–(5) as:

$$\Delta P = \frac{8\sqrt{3}QR^2\tau^2\mu L}{A\pi r^4} \tag{6}$$

The details of all the parameters in the formula are shown in Table A1.

The injection pressures of different solutions at different injection flow rates are then calculated and compared with the measured values as shown in Figure 13.

The measured pressures are generally higher than the calculated ones, and yet their changing trends are similar. The difference between the measured and theoretical values are possibly due to external factors, such the error of each experimental assembly and the measurement error of pressure sensor. In addition to the above errors, the reason for the large difference in the experiment is that the pulverized coal is more concentrated in the crack. Due to the fast flow rate, the pulverized coal accumulates in the channel, and the pressure peak rises, resulting in a large deviation between the test pressure and the theoretical pressure. The standard deviation of the test is shown in Table A2.

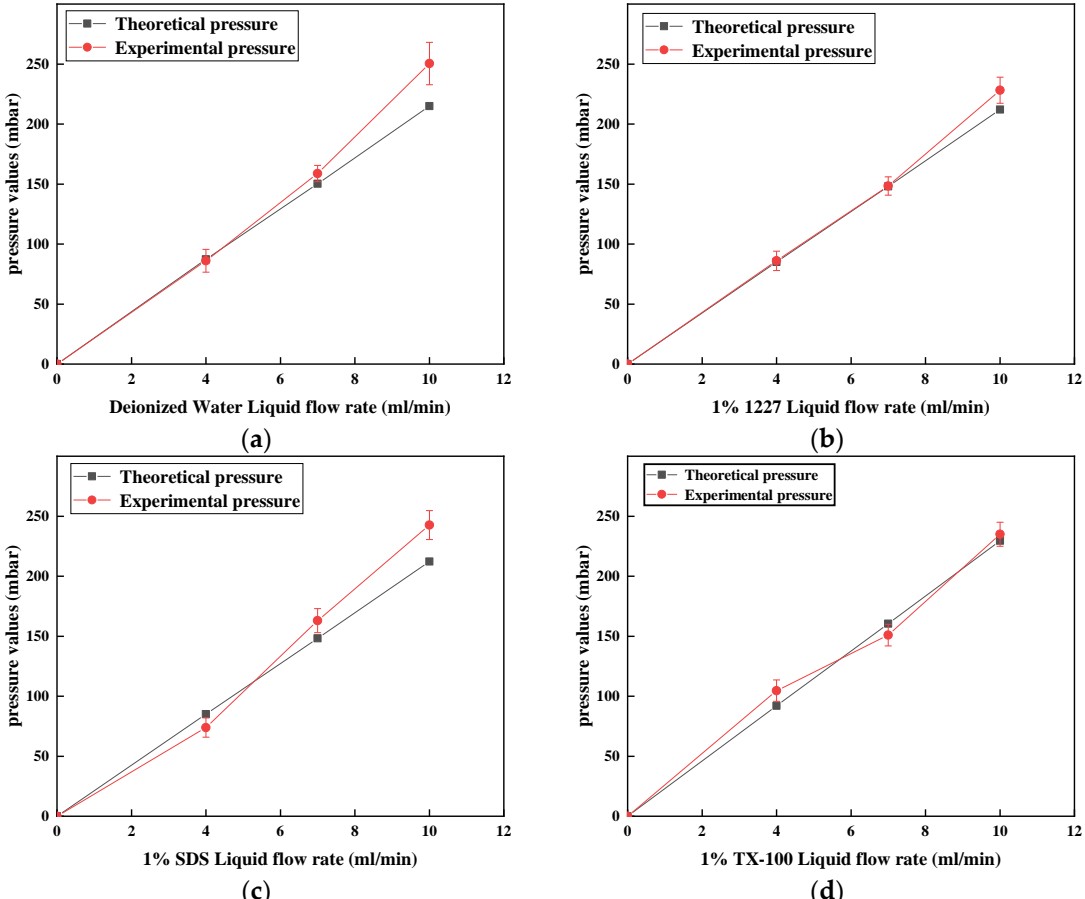

**Figure 13.** Comparison of theoretical and measured injection pressures. (**a**) Pressure of deionized water. (**b**) Pressure of 1% 1227. (**c**) Pressure of 1% SDS. (**d**) Pressure of 1% TX-100.

## 4. Conclusions

In the present work, the seepage of hydraulic fracturing fluid in a propped fracture was visualized for different fluids at different injection flow rates. The corresponding inlet pressure changes were analyzed, and the ratios of pulverized coal under different conditions were obtained from the fracture images to evaluate the blockage degree of propped fracture. Based on the results, the following conclusions can be drawn.

(1) All the inlet pressures of water and the solutions of three surfactants gradually increase with the increase of injection flow rate. The pulverized coal flows with the fluid. Some of the coal particles accumulate around the proppant and block the flow channel, while some find new flow channels in the propped fracture and flow along with the fluid until they are discharged. Through experimental observation, the above conclusions are basically reflected in each solution and each flow rate. The difference is that there are more blockages in deionized water. This blockage phenomenon is a common one. There is no clear standard, but it can be observed by the change in pressure and image analysis to determine whether there is a blockage. The blocked flow channel may be washed through by the subsequent flow, which causes fluctuations of inlet pressure. When the flow rate, proppant and coal powder reach dynamic equilibrium, the inlet pressure becomes stable. This stability occurs for a long time after the injection of the liquid. In proppant cracks, as the injection time increases, the pulverized coal changes with the flow of the liquid. Continue to inject the liquid until the pressure does not change significantly. The dynamic balance of flow rate, proppant, and pulverized coal is formed.

(2) Different surfactants reduce the inlet pressure differently. The solutions of all three different surfactants injected with coal powder into propped fractures show lower injection

pressures. Compared to that of deionized water, the injection pressure of SDS solution at the low flow rate drops by 14.2%, and those of the TX-100 solution and 1227 solution drop more obvious at the high flow rate, by 6.2% and 8.9%, respectively. Both anionic and cationic surfactants can alter the surface electrical properties of the coal powder, which reduces the accumulation and deposition of pulverized coal in the fracture and thus prevents the seepage channel blockages.

**Author Contributions:** Conceptualization, L.Y. and Q.H.; methodology, Q.Z. and Q.H.; software, Z.H.; validation, E.W. and Z.H.; formal analysis, L.Y.; investigation, Q.Z.; resources, Q.H.; data curation, Z.H.; writing—original draft preparation, L.Y.; writing—review and editing, Q.Z.; visualization, E.W.; supervision, Q.H. and L.Y.; project administration, Q.Z. and L.Y. All authors have read and agreed to the published version of the manuscript.

**Funding:** This research was funded by the National Natural Science Foundation of China (Grant No. 51974176, 52174194), the Shandong Province Natural Science Foundation of Outstanding Youth Fund (Grant No. ZR2020JQ22).

**Data Availability Statement:** Date are contained within the article.

**Conflicts of Interest:** The authors declare no conflict of interest.

## Appendix A

**Table A1.** The formula parameter table in this paper.

| | | |
|---|---|---|
| Q | The volume of liquid injected | 4 mL/min = 0.067 cm$^3$/s<br>7 mL/min = 0.117 cm$^3$/s<br>10 mL/min = 0.167 cm$^3$/s |
| A | Fracture cross-sectional area | 0.7 cm$^2$ |
| L | Crack length | 3 cm |
| $\mu$ | The viscosity of the liquid | SDS (0.00098 pa·s)<br>1227 (0.00102 pa·s)<br>TX-100 (0.00106 pa·s) |
| r | Effective flow radius | 0.002634 $R^4 = r^4$ |
| R | The average radius of proppant particle size | 1.105 mm |
| $\tau$ | The tortuosity of effective flow pores | 5.08 |
| $\pi$ | The ratio of circumference to diameter | 3.14 |
| $\Delta P$ | The pressure difference between the inlet and the outlet | |

**Table A2.** Experimental standard deviation table.

| NO. | Fluid | Error |
|---|---|---|
| 1 | water | 9.5 |
| 2 | water | 6.8 |
| 3 | water | 17.6 |
| 4 | 1% SDS | 8.1 |
| 5 | 1% SDS | 10 |
| 6 | 1% SDS | 12 |
| 7 | 1% 1227 | 8 |
| 8 | 1% 1227 | 7.6 |
| 9 | 1% 1227 | 10.8 |
| 10 | 1% TX-100 | 8.9 |
| 11 | 1% TX-100 | 9 |
| 12 | 1% TX-100 | 10 |

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
