# Peer review of "Experimental Study on Migration and Intrusion Characteristics of Pulverized Coal in Propped Fractures"

_processes, doi:10.3390/pr11072074_

Round 1

Reviewer 1 Report

In this paper, the research ideas are clear, the research methods are accurate, and the experimental steps are complete. Provide reference and basis when using surfactants to improve the water injection effect of coal seams. However, in order to make the paper better published and more readable for readers, I asked the following questions, which I hope the author will revise.

1. What is the basis for the selection of surfactants?

2. Select 120 mesh pulverized coal for the test, whether to consider selecting other mesh pulverized coal for the test?

3.In the analysis of Chapter 2.1, what is the formula for calculating the pressure growth rate of other solutions by using the pressure of water as the reference pressure?

4. How to control the crack opening in the experimental equipment?

Minor editing of English language required

Author Response

Dear Reviewer: 

Thank you for your effort and consideration for possible publishing our manuscript entitled "Experimental study on migration and intrusion characteristics of pulverized coal in propped fractures". All the raised comments are all valuable and very helpful for improving our manuscript. We have carefully gone through all the raised comments and we addressed all the comments point-by-point which improves the quality of the revised manuscript. 

Thanks,

Qiming Huang, Ph.D

College of Safety and Environmental Engineering

Shandong University of Science and Technology

Reviewer 2 Report

The article concerns to very important problem of reducing coal dust by applying of different liquids (water and chemical compounds). The work is based on experimental test which help to found dependence between inlet pressure and flow rate. This work can be used by the chemical industry in the field of environmental protection. However, the work requires some explanations:

11.       The article shows variation of pressure grow rate for different fluids. How were the pressure grow rates calculated? The pictures show almost the same inlet pressure after 5 min for all flow rates and liquids.

22.       It is known from hydraulic theory that for higher flow rates the inflow pressure should be higher. Please explain the reason of different pressure drops and bed permeability for various liquids.

33.        It would be advisable to provide the physical properties of the liquids used.

44.       How to explain such big difference of value R for TX100 shown in Fig. 8 and 9 for 7 and 10 ml/min?

55.       Figure 10 presents the same results as previous figures. Please justify such presentation.

66.       Line 203 – a reference should be given at name Carman-Kozeny. Have the formulas given from literature? If yes, the literature should be posted.

77.       On the obtained results the conclusions or discussion should contained suggestions for applications and indicate which method is recommended.

88.       The paper includes almost only the Chinese references. Has no one else in the world dealt with this problem?

The paper requires small changes and explanation given in this review for better understanding by potential readers. After correction the paper can be published in the Journal.

Author Response

(The authors gave the same response as above.)

Reviewer 3 Report

The authors studied migration and intrusion characteristics of pulverized coal in propped fractures. Effects of different parameters such as type of surfactant and flow rate are investigated systematically. There are some comments to enhance the quality of the paper:

1-    It is beneficial to add information of surfactants (manufacturer, purity, etc.) in the experimental design section.

2-    Line 111: There is a type in the sentence: “Compared with tthat of deionized water…”

3-    Lines 110-120: The authors suggest that several mechanisms (channelling and wetting) occur with behaviour of SDS surfactant at different fluid velocities. Is there proof in your experiments to support your claims? If there is not any tests, a detailed explanation with references is required.

4-    At the end of section 3.1, It is useful to include a paragraph summarizing the behaviour and reasons for the differences in performance of 3 surfactants.

5-    It is useful to include references for the equations 2-6. 

As a typo was found in your manuscript. It is suggested to do a fresh English review of the paper.

Author Response

(The authors gave the same response as above.)

Reviewer 4 Report

Please see my critical comments in the attached PDF file. Once you decide to rework the manuscript, I expect you to correct each highlighted part of the text and provide concise replies to each comment right in a PDF file using the Reply to Comment functionality.

Please do not answer in the comment, but rather add the answers to the manuscript's text.

carefully check and revise the spelling errors in the text of the manuscript

Author Response

Dear Reviewer: 

Thank you for your effort and consideration for possible publication of our manuscript entitled "Experimental study on migration and intrusion characteristics of pulverized coal in propped fractures". All the comments raised are  valuable and very helpful for improving our manuscript. We have carefully gone through all the raised comments and used the ' comment reply ' function to reply to the highlighted markup part in the PDF file. Detailed modifications will be added to the manuscript.

Thanks,
Qiming Huang, Ph.D
College of Safety and Environmental Engineering
Shandong University of Science and Technology

Round 2

Reviewer 4 Report

Please see my critical comments in the attached PDF file. Once you decide to rework the manuscript, I expect you to correct each highlighted part of the text and provide concise replies to each comment right in a PDF file using the Reply to Comment functionality.

Please do not answer in the comment, but rather add the answers to the manuscript's text.

Please see my critical comments in the attached PDF file. Once you decide to rework the manuscript, I expect you to correct each highlighted part of the text and provide concise replies to each comment right in a PDF file using the Reply to Comment functionality.

Please do not answer in the comment, but rather add the answers to the manuscript's text.

Author Response

Dear Reviewer: 

Thank you for your effort and consideration for possible publication of our manuscript entitled "Experimental study on migration and intrusion characteristics of pulverized coal in propped fractures". We have carefully gone through all the raised comments and used the ' comment reply ' function to reply to the highlighted markup part in the PDF file. The second revision of the manuscript is marked in red font and yellow highlight.

Thanks,
Qiming Huang, Ph.D
College of Safety and Environmental Engineering
Shandong University of Science and Technology

Round 3

Reviewer 4 Report

Please see my critical comments in the attached PDF file. Once you decide to rework the manuscript, I expect you to correct each highlighted part of the text and provide concise replies to each comment right in a PDF file using the Reply to Comment functionality.

Please do not answer in the comment, but rather add the answers to the manuscript's text.

Author Response

Dear Reviewer: 

Thank you for your effort and consideration for possible publication of our manuscript entitled "Experimental study on migration and intrusion characteristics of pulverized coal in propped fractures". We have carefully gone through all the raised comments and used the ' comment reply ' function to reply to the highlighted markup part in the PDF file. The third revision of the manuscript is marked in green font.

Thanks,
Qiming Huang, Ph.D
College of Safety and Environmental Engineering
Shandong University of Science and Technology
